# Antidepressant Activity of Agarwood Essential Oil: A Mechanistic Study on Inflammatory and Neuroprotective Signaling Pathways

**DOI:** 10.3390/ph18020255

**Published:** 2025-02-14

**Authors:** Shunan Zhang, Xiqin Chen, Canhong Wang, Yuanyuan Sun, Bao Gong, Dan Li, Yulan Wu, Yangyang Liu, Jianhe Wei

**Affiliations:** 1Hainan Provincial Key Laboratory of Resources Conservation and Development of Southern Medicine & Key Laboratory of State Administration of Traditional Chinese Medicine for Agarwood Sustainable Utilization, Hainan Branch of the Institute of Medicinal Plant Development, Chinese Academy of Medical Sciences and Peking Union Medical College, Haikou 570311, China; zhangshunan2000@gmail.com (S.Z.); acxgdxq@163.com (X.C.); xinzhuangjianpo@163.com (C.W.); syy2742132993@163.com (Y.S.); gongbao0112@aliyun.com (B.G.); wyl190903008@163.com (Y.W.); 2Key Laboratory of Bioactive Substances and Resources Utilization of Chinese Herbal Medicine, Ministry of Education & National Engineering Laboratory for Breeding of Endangered Medicinal Materials, Institute of Medicinal Plant Development, Chinese Academy of Medical Sciences and Peking Union Medical College, Beijing 100193, China; 3School of Biological and Food Engineering, Guangdong University of Petrochemical Technology, Maoming 525011, China; 4The Burdon Sanderson Cardiac Science Centre and BHF Centre of Research Excellence, Department of Physiology, Anatomy and Genetics, University of Oxford, Oxford OX1 3PT, UK; dan.li@dpag.ox.ac.uk

**Keywords:** agarwood essential oil, aromatherapy, antidepressant, inflammatory, neuroprotective, NF-κB/IκB-α pathway, BDNF/TrkB/CREB pathway

## Abstract

**Background**: Depression ranks among the most severe mental health conditions, and poses a burden on global health. Agarwood, an aromatic medicinal plant, has shown potential for improving mental symptoms. As a common folk medicine, agarwood has been applied as an alternative method for mental disorders such as depression through aromatherapy. Previous studies have found that the therapeutic effects of agarwood aromatherapy are primarily related to its volatile components. This study aimed to examine the antidepressant properties and underlying mechanisms of agarwood essential oil (AEO), a collection of the volatile components of agarwood utilized through aromatherapy inhalation and injection administration in mice. **Methods:** A lipopolysaccharide (LPS)-induced inflammatory depression model was used to evaluate the effects of AEO inhalation and injection on depression-like symptoms. Behavioral assessments included the open-field, tail suspension, and forced swimming tests. Western blot (WB) and ELISA techniques were used to further verify the mechanistic insights. **Results:** In the LPS-induced depression-like model, AEO inhalation and injection significantly improved depression-like symptoms, decreased immobility duration in both the tail suspension and forced swimming tests in model mice, and reduced the levels of inflammatory cytokines IL-1β, IL-6, and TNF-α. WB experiments demonstrated that AEO inhibited the NF-κB/IκB-α inflammatory pathway and activated the BDNF/TrkB/CREB pathway in the hippocampus of the LPS-depression model mice. Notably, AEO extracted by hydrodistillation was more effective in alleviating LPS-induced depressive-like behaviors than using supercritical CO_2_ fluid extraction. **Conclusions:** Both the inhalation and the injection administration of AEO exerted notable antidepressant effects, potentially associated with reducing inflammation levels in the brain, downregulating inflammatory NF-κB/IκB-α, and upregulating the neuroprotective BDNF/TrkB/CREB signaling pathway. In the future, it is necessary to further determine the pharmacodynamic components, key targets and specific molecular mechanisms of AEO’s antidepressant effects so as to provide more support for the neuroprotective research of medicinal plants.

## 1. Introduction

Depression is a widespread and prevalent mental disorder, characterized by symptoms such as a persistently low mood, lowered will and slowed thinking, which can potentially trigger suicidal behavior [1]. The 2019 Global Burden of Disease, Injury, and Risk Factors (GBD) study showed that depression is progressively becoming more prevalent, ranking amongst the most disabling of mental disorders and frequently associated with serious co-morbidities [2]. Since the onset of the COVID-19 pandemic, numerous factors contributing to poor mental health have increased, leading to a continued rise in the number of people experiencing depression [3]. The causes of depression are multifaceted and diverse. The cytokine hypothesis and the neuroplasticity functional hypothesis have both been put forth recently, in addition to the widely accepted monoamine hypothesis, related to neuroendocrine and neuro-immune disorders [4,5]. Clinical and research data have reported that inflammatory markers in both the blood and the brain, including interleukin IL-1β, IL-6, and tumor necrosis factor TNF-α, are markedly elevated in individuals with major depression. Inflammatory cytokines could induce and promote the development of depression through multiple neurological pathways [6,7,8]. Several studies have also confirmed that neuroplasticity dysfunction is an important mechanism in depression, with structural alterations in brain regions such as the hippocampus and prefrontal cortex in depressed patients being closely linked to the decreased secretion of brain-derived neurotrophic factor (BDNF) [9,10]. In recent years, many phytochemicals in natural products have gradually been found to have beneficial effects in alleviating and treating depression [11].

Agarwood (*Aquilaria sinensis* (Lour.) Spreng.) has been used for centuries, and is widely recognized for its potential therapeutic benefits. The *Compendium of Materia Medica*, written by Li Shizhen, describes agarwood as having the properties of excitement, wind-repelling, aphrodisiac, anti-rheumatic, anti-malarial, analgesic, laxative, nourishing and diuretic effects [12]. In traditional Chinese medicine, it is utilized as an aphrodisiac, sedative, cardiotonic, and carminative, and is employed to address stomach issues, coughs, rheumatism, and high fever [13]. In Thailand, agarwood has traditionally been used to treat various infectious diseases, including diarrhea and skin conditions [13]. In traditional Arabic medicine, agarwood is frequently used to address neurodegenerative diseases, digestive system disorders and sedative diseases [14]. Common folk uses of agarwood include burning it in incense sticks and using its essential oils for aromatherapy. Burning incense sticks produces aromatic components that not only calm and soothe people, but also have the effect of repelling insects [13]. In addition, heating the incense sticks and allowing mice to inhale the volatile gases of agarwood through their mouths and noses alleviated the learning and memory deficits induced by scopolamine, confirming the aromatic and cognitive-enhancing effects of agarwood [15]. The essential oil of agarwood, a collection of its volatile components, exhibits a range of pharmacological activities, including soothing the body and mind, regulating the mood, calming nerves, and providing antibacterial and anti-inflammatory effects [16,17]. It is very popular among consumers in the Middle East, especially in Southeast Asia. Aromatherapy using agarwood essential oil is a natural method of treatment for depression, with recognized efficacy and mechanisms. It not only has a history of application and relevant records among people, but it also shows good anti-depressant effects in modern research. The yield of AEOs obtained using the conventional water steam distillation process (SAEO) is typically between 0.1% and 0.5%. Although supercritical CO_2_ fluid extraction (CAEO) can increase the yield to as much as 0.8% to 5%, many essential oil enthusiasts are uncomfortable with this extraction method [18]. Studies have shown that the AEO obtained from these two approaches differed significantly in terms of aroma, chemical composition, and antioxidant, antibacterial and anti-inflammatory effects [19,20]. However, the fundamental reasons for the differences in composition and activity have not been fully elucidated, and the differences in their efficacy in terms of neuro-modulation, such as antidepressant effects, have not been reported.

The group’s previous study showed that AEO has significant pro-sleep, anti-anxiety and anti-depressant effects, and its anti-depressant effect is related to inhibiting the excessive activity of the hypothalamic–pituitary–adrenal (HPA) axis (significantly inhibiting nNOS protein levels in the hippocampus and reducing cerebral cortex and hippocampal corticotropin-releasing factor gene expression), and regulating the high sensitivity of the HPA axis as well as the content of monoamines and cholinergic neurotransmitters, such as 5-hydroxytryptamine (5-HT), γ-aminobutyric acid (GABA) and glutamate (Glu), in the brain [21,22,23]. Additionally, several studies have shown that AEO and its compounds have significant neuroprotective effects on neuroinflammation and hippocampal oxidative damage, and support BDNF levels in the brain [24,25,26,27]. However, the exact mechanism by which AEO exerts its antidepressant effects through anti-inflammatory pathway and neurorepair remains unclear. Activating the BDNF/TrkB/CREB signaling pathway or regulating pathway-related proteins can increase the content of monoamine neurotransmitters, promote the activation of growth factors, and protect synaptic plasticity, thereby producing an antidepressant effect [28,29]. Therefore, this study attempted to link the pharmacological activity of AEO to depression through this signaling pathway.

In this study, we utilized a classical inflammatory depression model induced by lipopolysaccharide (LPS) to mimic high inflammation levels and the depression-like behaviors characteristic of major depression in vivo [30,31,32]. Behavioral tests were conducted to assess the antidepressant efficacy of AEOs obtained from two extraction methods (SAEO and CAEO) of the same agarwood. Additionally, we investigated NF-κB and BDNF pathway-related proteins in the hippocampal tissues of LPS-induced mice, aiming to explore the underlying mechanism of action of AEO. This study not only fills the gap in research regarding the differences in active components and antidepressant activity between two types of AEO, but also advances the understanding of AEO’s neuroregulatory effects, specifically its antidepressant potential. By linking AEO’s pharmacological activity to antidepressant mechanisms through signaling pathways, this research contributes new insights into the broader applications of AEO in neuroprotection and mood disorders.

## 2. Results

AEO and positive drugs, such as paroxetine, were administered for 7 consecutive days. Except for the control group, which was injected with the corresponding volume of normal saline, all mice were intraperitoneally injected with 1 mg·kg^−1^ LPS 1 h after the administration of AEO and positive drugs, starting from the 5th day. Two to six hours after LPS intraperitoneal injection, the mice developed acute and severe symptoms, such as generalized shivering, elevated body temperature, reduced feeding, the massive excretion of unformed feces, and rapid weight loss, consistent with documented findings in the literature. After 24 h, mice gradually recovered their voluntary activity behavior; however, during the tail suspension test (TST) and forced swimming test (FST), there was an increase in immobility and signs of depression [33].

### 2.1. SAEO Attenuates LPS-Induced Depression-like Behavior in Mice

A comparison of the LPS model group with the control group is shown in Figure 1A–D and Table 1. The mice lost significant body weight in the LPS model group, while the AEO inhalation and injection groups and positive drug group showed effectively reduced weight loss and alleviated disease-like symptoms. In the open-field test (OFT), all mice resumed voluntary activity 24 h after the injection of LPS intraperitoneally. There was not significant difference in the total distance traveled over 5 min between the control and model groups, indicating that LPS did not affect the voluntary activity of mice. This suggests that a measure of mice’s immobility time during subsequent TSTs and FSTs was not due to physical weakness. In TST and FST, a significant increase in immobility time was observed in the model group. In both the AEO administration group and the positive drug group, immobility time was reduced in mice induced with LPS, and the difference was statistically significant according to one-way analysis of variance (ANOVA). This experiment demonstrates, for the first time, that both AEO aromatherapy and injection administration can exert a rapid anti-LPS inflammatory depressive effect, further confirming the antidepressant activity of AEO.

### 2.2. SAEO Has Stronger Antidepressant Activity than CAEO

Previous studies have shown that CAEO and SAEO differ in chemical composition, as well as antioxidant and anti-inflammatory capacity. However, no studies have examined the underlying reasons for the difference in popularity between the two in the marketplace. Apart from usage habits, the difference in the real-life effectiveness of the two oils may be a primary factor. AEO has a calming effect and is often used in aromatherapy to assist in the treatment of some mental illnesses, such as insomnia, anxiety, depression, etc. [19]. Therefore, we compared the rapid antidepressant effects of two essential oils produced by different extraction methods from the same source. The specific experiments are shown in Figure 1E,F and Table 2, and the experimental doses are provided according to the literature and the previous experience of the group. In both TST and FST, the model mice showed a significant increase in immobility time compared to the control group, and both SAEO administration groups reduced the mice’s immobility time (Figure 1E,F). Although the CAEO administration group showed alleviate depressive symptoms in mice to some extent, only high-dose CAEO incense was effective in reducing mice’s immobility time during TST and FST. According to the test results, SAEO was more effective than CAEO in alleviating inflammatory depression in LPS at the same essential oil dose, either by heated aromatherapy or injected administration.

### 2.3. SAEO Reduces LPS-Induced Inflammation in Mice

In this experiment, TNF-α, IL-6, and IL-1β levels were significantly increased in the serum and cortical supernatant of the LPS model mice compared with the control group (Figure 2, Table 3), while in LPS-induced mice, SAEO reversed pro-inflammatory cytokine increase to varying degrees, and had a significant anti-inflammatory effect.

### 2.4. SAEO Blocks the LPS-Induced Expression of NF-κB and IκB-α in the Hippocampus of Mice

Studies have shown that LPS administration produces peripheral and neuro-inflammation, causing the activation of nerve cells and ultimately a decrease in neurotransmitters, neurotrophic factors, etc. [34]. At the same time, the role of the hippocampus in regulating depression-related signaling pathways has been widely studied, including signaling pathways controlled by upstream factors such as NF-κB and BDNF [35]. The study found that the content of BDNF mRNA was highest in the hippocampus, followed by the cerebral cortex [36]. To examine the neuroprotective benefits of SAEO, its effects on the expression of NF-κB/IκB-α and BDNF/TrkB/CREB pathway-related proteins in the hippocampal tissues of LPS-induced mice were measured.

As shown in Figure 3 and Table 4, WB analysis revealed a significant increase in the expression of *p*-NF-κB and *p*-IκB-α proteins in the hippocampal tissues of LPS-induced mice compared to the control group, which is consistent with reports in the literature [37]. The mouse hippocampal NF-κB/IκB-α pathway was activated after LPS intraperitoneal injection, indicating that LPS induced neuro-inflammation in mice. Both paroxetine and SAEO treatment significantly reversed inflammation-associated protein expression and significantly inhibited the LPS-induced phosphorylation of NF-κB and IκB-α proteins in the mouse hippocampus.

### 2.5. SAEO Increases LPS-Induced CREB/BDNF Expression in the Mouse Hippocampus

BDNF is the most prevalent neurotrophic factor in the body and plays a crucial role in regulating the selective survival of specific neuronal populations during development and in differentiation [38,39]. Figure 4 and Table 5 demonstrates that LPS injection notably decreased the expression levels of BDNF, its receptor TrkB and the regulatory protein CREB in mouse hippocampal tissues compared with the control group, while paroxetine reversed this effect. Additionally, medium and high doses of SAEO aromatherapy and 40 mg/kg essential oil injection increased BDNF, TrkB and CREB protein expression levels to different degrees compared with the LPS group.

## 3. Discussion

Based on the neuroprotective effects of incense and its essential oil, this study sought to reassess the effects of incense essential oil on mice with inflammatory depression, and explore its mechanism of action. To better reflect the clinical and real-life use of AEO, this experiment was designed for both heated aromatherapy and injectable administration, with doses selected based on previous experiments and experience [22,23]. A comparison of the antidepressant efficacy of AEO obtained through different extraction methods of the same source of agarwood was also evaluated. The results of behavioral tests show that both SAEO aromatherapy and injectable administration significantly alleviated depressive behavior, demonstrating effects similar to those of paroxetine. However, the antidepressant effect of CAEO was less potent than that of SAEO at the same dose, probably due to differences in their composition. This variation may help explain why CAEO is less popular than SAEO in the market. Interestingly, in the previous study, we found that in terms of chemical composition, sesquiterpene compounds, the main component of SAEO, accounted over 80% of its content, whereas the total sesquiterpene content in CAEO was only 62%. Additionally, SAEO had more significant anti-inflammatory effects than CAEO [18]. Therefore, we conjecture that the antidepressant component of AEO is likely to be represented by the unique sesquiterpene compounds in agarwood. These compounds are small, volatile, lipid-soluble molecules with a relative molecular weight of less than 300 Da, which can easily cross the blood–brain barrier and play a protective role against brain diseases. Previous molecular docking results have shown that these sesquiterpene components in agarwood can strongly bind to neurotransmitter receptors in the brain to exert neuromodulatory effects [22].

The development of depression is accompanied by an increase in inflammation. LPS binds to Toll-like receptors (TLRs) on the cell membrane surface, which undergo conformational changes in their intramembrane motifs, transferring signals to the interior of immune cells and activating pathways such as NF-κB associated with inflammation through a series of signaling events [34]. In most normal cells in a quiescent state, NF-κB binds to its inhibitory protein IκB, etc. The classical activation pathway mainly involves the phosphorylation of IκB by IκB kinase, which derepresses NF-κB. Activated NF-kB induces the transcription of inflammation-related genes, leading to the release of pro-inflammatory cytokines, including TNF-α and IL-6, and promoting peripheral and central inflammation [32,40,41,42]. The NF-κB/IκB-α pathway can exert anti-inflammatory effects, reduce the content of inflammatory factors, improve oxidative stress, activate neurotrophic factors, and effectively reverse LPS-induced depressive-like behavior [32]. Several studies have shown that a rapid increase in inflammatory factors in the peripheral serum and brain tissue in mice is one of the most important features of LPS intraperitoneal injection, an index used to model the increase in inflammatory factors in depressed patients, and the mRNA and protein expression levels of TNF-α, IL-6, and IL-1β were significantly increased in the cortices of suicidal patients with depression [43,44,45]. Furthermore, during the acute phase, when cytokine levels were elevated, levels of TNF-α and IL-6 were highest in the cortex [46]. In the present study, SAEO was found to attenuate the increases in TNF-α, IL-6, and IL-1β in peripheral and central mice in vivo in LPS-induced mice. To verify the effect of SAEO on the LPS-activated NF-kB pathway, we examined the expression of phosphorylated and non-phosphorylated NF-kB p65 and IκB-α proteins in mouse hippocampal tissues. The experimental results show that SAEO has a similar effect to paroxetine in blocking NF-κB/IκB-α pathway activation.

BDNF plays a crucial role in neuronal survival, development and synaptic plasticity [38]. Increasing evidence indicates that BDNF levels are diminished in individuals with depression, and that the BDNF concentrations in peripheral blood are closely associated with the severity of the condition [47,48]. It is now known that the direct infusion of BDNF into the brain restores astrocyte immunoreactivity in the hippocampus of depressed rats, demonstrating antidepressant potential [49]. In contrast, LPS modeling mimics the symptoms of neuronal cell death, reduced neurogenesis and reduced expression of BNDF in the brains of depressed patients [50]. Therefore, raising the low BNDF in patients is also one of the new strategies for treating depression that are currently receiving significant attention [51]. Studies have shown that the behavioral effects of antidepressants require BDNF signaling through TrkBy activated by TrkB, and that blocking this pathway can lead to the failure of the antidepressant ketamine [52]. In addition, BDNF is a target gene of CREB, and the increased activation of CREB enhances the expression of BDNF, thereby stimulating the synthesis of endogenous BDNF [53]. The upregulation of the BDNF/TrkB/CREB signaling pathway, accompanied by the activation of TrkB and growth factors, can significantly reduce depressive-like behaviors and alleviate depression-induced microscopic changes in the frontal cortex and hippocampus [28]. In this study, we assessed the impact of SAEO on the reduced expression of BDNF/TrkB/CREB pathway proteins induced by LPS. The results demonstrate that SAEO effectively reversed these reductions, suggesting that this pathway may contribute to its antidepressant effects (Figure 5). Our results are consistent with the results of several other studies investigating the neuroprotective and antidepressant properties of essential oils. For example, the monoterpenoid essential oil Alpha-pinene can alleviate memory impairment by restoring BDNF/TrkB/CREB in the rat hippocampus [54]. Saffron essential oil improves the depressive-like behavior of mice by affecting the release of neurotransmitters in serum, upregulating the levels of BDNF, etc., and improving hippocampal neuronal damage [55].

## 4. Materials and Methods

### 4.1. Drugs and Animals

The raw material used for AEO extraction was the same batch of agarwood subjected to the whole-tree agarwood-inducing technique (batch number: CXT20161009-1), and the agarwood samples were identified by researcher Yangyang Liu and kept at the Analysis and Identification Center of Hainan Branch of the Institute of Medicinal Plant Development, Chinese Academy of Medical Sciences. The hydrodistilled agarwood essential oil (SAEO) was homemade in the laboratory; 1 kg was weighed and placed in a 10 L round-bottom flask of agarwood, and 7.5 L of water was added to heat and reflux for extraction. The supercritical CO_2_ extracted agarwood essential oil (CAEO) was entrusted to Guangdong Yufeng Chenxianghui Technology Co., Ltd. (Maoming, China) for extraction; 5 kg of agarwood was weighed and extracted using a supercritical CO_2_ extractor, with an extraction temperature of 60 °C, an extraction pressure of 26 MPa, an extraction time of 2 h, and a supercritical CO_2_ flow rate of 260 L·h^−1^. The extract was molecularly distilled and purified to remove excess fatty acid components. Both were recognized by GC-MS chemical analysis. While the overall sesquiterpene content of CAEO was only 62% and there were 17% of 2-(2-phenylethyl) chromones, the total sesquiterpene content of SAEO was up to 84% [18]. LPS (*Escherichia coli*, O55:B5) was purchased from Sigma Aldrich Trading Co., Ltd. (Shanghai, China), and paroxetine was purchased from AccuStandard (New Haven, CT, USA).

The tail suspension/forced swim apparatus (RD1119-FSTS-M-4) was purchased from Shanghai Xinruan (Shanghai, China). The OFT chamber (35 cm × 35 cm × 30 cm, L × W × H) (RD1118-CO-M-4) was purchased from Shanghai Xinruan (Shanghai, China). The electronic aromatherapy oven was purchased from Xixiangzhuan, (Quanzhou, China). The Multiskan GO microplate spectrophotometer was purchased from Thermo Fisher Scientific (Waltham, MA, USA). The TGL-16M benchtop high-speed refrigerated centrifuge was purchased from Shanghai Luxiang Instrument Co., Ltd. (Shanghai, China). The ChemiDoc XRS + chemiluminescence imaging system was purchased from Bio-Rad Laboratories (Hercules, CA, USA).

The 48 SPF adult male Balb/c mice, of weight 18–22 g, were provided by Hunan Slaughter Jingda Laboratory Animal Co., Ltd. (Changsha, China, production license: SCXK(Xiang)2019-0004). All animals were fed and watered ad libitum and kept under a 12 h light–dark cycle with light hours of 8:00–18:00, and the temperature of the feeding environment was maintained at 21–25 °C. All animal experiments were performed in accordance with laboratory animal care and guidelines, and the animal experiments were approved by the Experimental Animal Welfare Ethics and Animal Experimentation Safety Committee of Hainan Pharmaceutical Research Institute Co., Ltd. (2022HL008). All procedures were designed to minimize animal suffering and distress. Anesthetic and analgesic protocols were strictly adhered to during invasive procedures. Animals were closely monitored for signs of distress or abnormal behavior, with any exhibiting significant discomfort or adverse effects promptly removed from the study and given appropriate veterinary care. Humane euthanasia was performed according to established protocols to minimize pain and distress.

### 4.2. Experimental Procedure

As shown in Figure 6, Balb/c mice were randomly divided into 6 groups (8 mice per group), as follows: control group, LPS group (1 mg/kg), LPS + SAEO group (4 μL), LPS + SAEO group (8 μL), LPS + SAEO intraperitoneal injection (ip.) group (40 mg/kg), and LPS + paroxetine group (10 mg/kg). The control group was injected with the corresponding volume of saline, and the LPS and positive groups were treated with intraperitoneal injections. Except for the AEO intraperitoneal injection group, all AEOs were administered by inhalation using an electronic aromatherapy oven heated at 80 °C for 1 h/d. The dose and mode of administration of essential oils were derived from the literature and previous studies of the subject group [22,23]. The specific experimental design is shown in Figure 6; AEO and positive drugs were administered for 7 consecutive days. Except for the control group, which was injected with the corresponding volume of normal saline, all mice were intraperitoneally injected with 1 mg·kg^−1^ LPS 1 h after the administration of AEO and positive drugs starting from the 5th day. After the behavioral test on the 8th day, the mice were executed and blood was collected; the serum was centrifuged at 4500 rpm for 15 min, and the serum was divided; the brain was quickly collected and rinsed with saline, the cortical and hippocampal tissues were stripped, and the serum and brain tissue were quickly stored at −80 °C.

#### 4.2.1. Behavior Test

##### Open-Field Test (OFT)

The mice were placed into the autonomic activity test chamber (35 cm × 35 cm × 30 cm) for 2 min after acclimation to start the test, and mouse movements were recorded for 5 min. During the test, the surrounding environment was kept quiet. VisuTrack 2.0 Animal Behavior Analysis Software was used to track the animal’s trajectory and calculate the distance traveled in the open-field experiment.

##### Tail Suspension Test (TST)

The mice were fixed with adhesive tape at 1 cm from the tail tip and hung upside down on a hook, so that they were suspended upside down, while the VisuTrack animal behavior analysis software automatically recorded the activity status of the mice within 6 min, and the last 4 min of tail suspension time was counted.

##### Forced Swimming Test (FST)

The mice were placed in a transparent cylinder (10 cm in diameter and 25 cm in height) with a water depth of 10 cm. The VisuTrack animal behavior analysis software automatically recorded the activity status of the mice during 6 min and calculated the cumulative immobility time of the mice during the latter 4 min.

#### 4.2.2. Enzyme-Linked Immunosorbent Assay (ELISA)

Serum and cortical tissue homogenate supernatant samples were stored at −80 °C for subsequent analysis. TNF-α, IL-1β and IL-6 levels in serum and cortical tissue homogenate supernatant samples were measured by use of ELISA kits (SenBeiJia Biological Technology, Nanjing, China).

#### 4.2.3. Western Blotting

Hippocampus tissue and tissue lysate were homogenized at a ratio of 1:9 (*w*/*v*) and centrifuged at 12,000 rpm for 15 min at 4 °C. The proteins obtained from the lysis were collected from the supernatant. The protein concentration was determined using the BCA kit. After adding loading buffer and lysate to unify the protein concentration, the protein was inactivated by boiling at 100 °C for 3 min. Protein extracts were separated by SDS polyacrylamide gel electrophoresis and then transferred to PVDF membranes. The membranes were closed with Fast Blocking Solution for 40 min at room temperature, washed with 1% TBST and incubated with the primary antibody at 4 °C overnight. We diluted the antibodies in commercial dilutions separately. The primary antibodies used were as follows: Phospho-NF-κB p65 (Cell signaling, 3033S, 1:1000, Danvers, MA, USA), Phospho-IκBα (Cell signaling, 2859S, 1:1000), Phospho-CREB (Cell signaling, 9198S, 1:1000), NF-κB p65 (Abcam, ab16502, 1:1000), IκB-α (Beyotime, AI096, 1:500, Shanghai, China), CREB (Beyotime, AF1018, 1:500), Phospho-TrkB (Beyotime, AF1963, 1:500), GAPDH (Beyotime, AF1186, 1:500), and BDNF (Affinity, DF8387, 1:500, Cincinnati, OH, USA). After washing with 1% TBST, the membrane was incubated with HRP-coupled secondary antibody (Cell Signaling, 1:2000) for 2 h at room temperature. Immunoreactive bands were displayed under a gel imaging system (ChemiDoc™ XRS +, Bio-Rad, Hercules, CA, USA) using chemiluminescent detection reagents after three washes with 1% TBST.

### 4.3. Statistical Analysis

All data in this paper were recorded as mean ± SD, and multiple groups were analyzed by one-way ANOVA using the GraphPad Prism 9.5.1 (GraphPad Software Inc., La Jolla, San Diego, CA, USA) software. Pairwise comparisons between groups were conducted using a post-hoc *t*-test. *p* < 0.05 was considered a statistically significant difference.

## 5. Conclusions

In this study, we demonstrated that the aromatherapy inhalation of AEO not only has a significant antidepressant effect comparable to injection, but it is also more practical for application due to its user-friendly form. Interestingly, at the same dose, SAEO has a better antidepressant effect than CAEO. The antidepressant effect of SAEO may be related to the alleviation of inflammation levels, the downregulation of the inflammatory NF-κB/IκB-α and the upregulation of the neuroprotective BDNF/TrkB/CREB signaling pathways in mice. However, further research is needed to identify the pharmacodynamic components, key targets and specific molecular mechanisms of AEO’s antidepressants effect. This study confirmed the antidepressant activity of agarwood essential oil, highlights its impact on inflammatory levels and neuroprotective signaling pathways associated with the antidepressant effect. While this study provides important insights into the antidepressant activity of AEO and its potential mechanisms, it is important to note that the findings are based on animal models, and the results should be validated in human studies. Future research will be needed to further explore the clinical applicability of AEO, particularly regarding its effectiveness and safety in human populations. This study lays a theoretical and experimental foundation for the safe use of agarwood essential oil, supports its potential use as a folk medicine therapy for depression, and contributes to neuroprotective research on medicinal plants.

## Figures and Tables

**Figure 1 pharmaceuticals-18-00255-f001:**
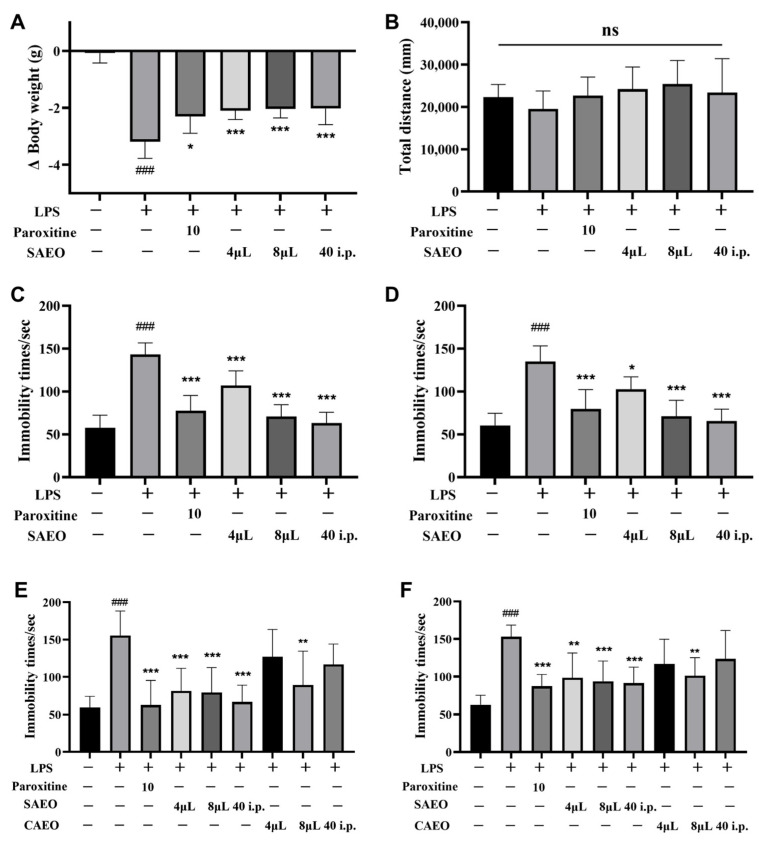
SAEO attenuates LPS model-induced depression-like behavior and two AEO anti-LPS-induced depressive activities in mice. (**A**) Body weight change of Balb/c mice from day 6 before LPS injection to day 8 before behavioral experiments. (**B**) Open-field experimental test in Balb/c mice. (**C**) Forced swimming experimental test in Balb/c mice. (**D**) Tail suspension experimental test in Balb/c mice. (**E**) Forced swimming experimental test in Balb/c mice (comparison of two AEO). (**F**) Tail suspension experimental test in Balb/c mice (comparison of two AEO). The data are expressed as the mean ± SD (*n* = 8). ### *p* < 0.001 compared to the control group; * *p* < 0.05, ** *p* < 0.01, *** *p* < 0.001 compared to the LPS-induced group, ns: non-significant.

**Figure 2 pharmaceuticals-18-00255-f002:**
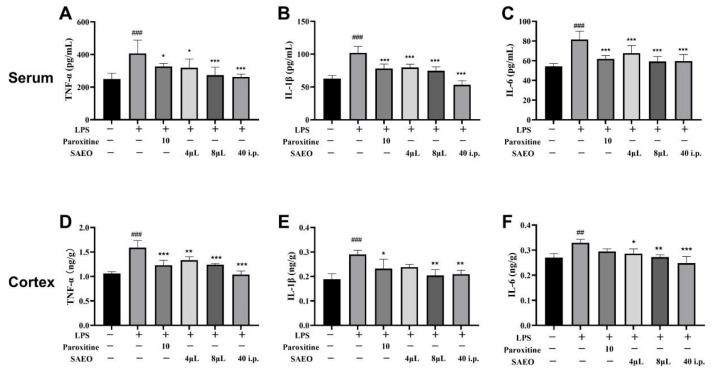
SAEO reduces LPS-induced inflammation levels in serum and cortical tissues of mice. (**A**) Changes in TNF-α levels in serum of Balb/c mice. (**B**) Changes in IL-1β levels in serum of Balb/c mice. (**C**) Changes in IL-6 levels in serum of Balb/c mice. (**D**) Changes in TNF-α levels in cortex of Balb/c mice. (**E**) Changes in IL-1β levels in cortex of Balb/c mice. (**F**) Changes in IL-6 levels in cortex of Balb/c mice. The data are expressed as the mean ± SD (*n* = 4–6). ## *p* < 0.01, ### *p* < 0.001 compared to the control group; * *p* < 0.05, ** *p* < 0.01, *** *p* < 0.001 compared to the LPS-induced group.

**Figure 3 pharmaceuticals-18-00255-f003:**
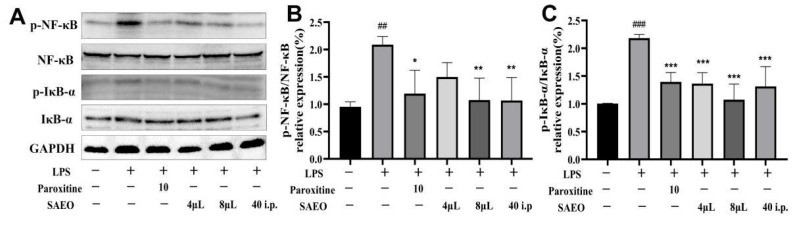
SAEO was able to inhibit the LPS-induced increase in NF-κB/IκB-α expression in the mouse hippocampus. (**A**) A WB map of NF-κB/IκB-α pathway proteins. (**B**) Quantitative results of *p*-NF-κB/ NF-κB relative protein expression levels in hippocampal tissues of Balb/c mice using Image J 1.51j8 software. (**C**) Quantitative results of *p*-IκB-α/ IκB-α relative protein expression levels in hippocampal tissues of Balb/c mice using Image J software. The data are expressed as the mean ± SD (*n* = 3–5). ## *p* < 0.01, ### *p* < 0.001 compared to the control group; * *p* < 0.05, ** *p* < 0.01, *** *p* < 0.001 compared to the LPS-induced group.

**Figure 4 pharmaceuticals-18-00255-f004:**
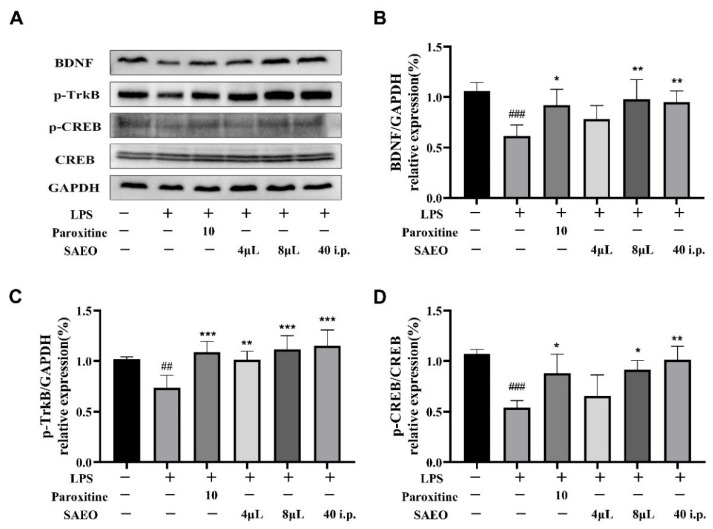
SAEO is able to reverse the LPS-induced reduction in BDNF/TrkB/CREB expression in mouse hippocampus. (**A**) WB map of BDNF/TrkB/CREB pathway proteins. (**B**) Quantitative results of BDNF/GAPDH relative protein expression levels in hippocampal tissues of Balb/c mice using Image J software. (**C**) Quantification of *p*-TrkB/GAPDH relative protein expression levels in the hippocampus of Balb/c mice using Image J software. (**D**) Quantitative results of *p*-CREB/CREB relative protein expression levels in hippocampal tissues of Balb/c mice using Image J software. The data are expressed as the mean ± SD (*n* = 3–5). ## *p* < 0.01, ### *p* < 0.001 compared to the control group; * *p* < 0.05, ** *p* < 0.01, *** *p* < 0.001 compared to the LPS-induced group.

**Figure 5 pharmaceuticals-18-00255-f005:**
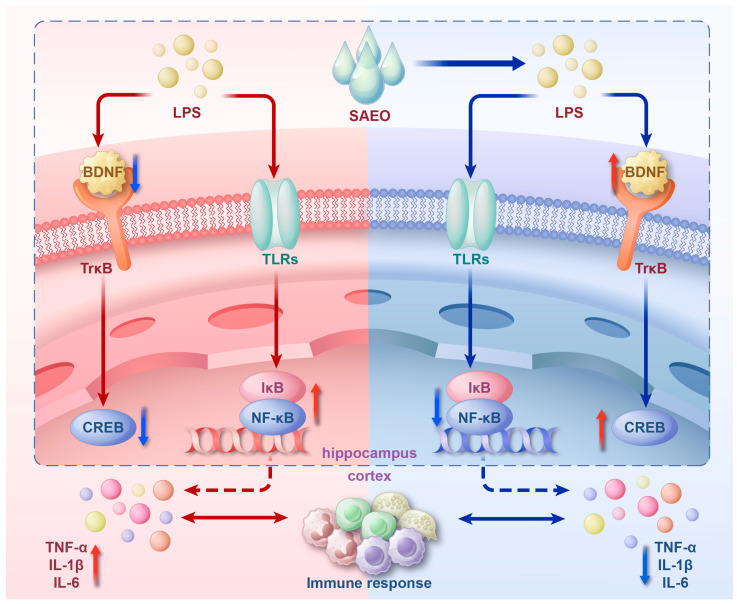
Schematic of SAEO regulating neuroinflammation and the neuroprotective mechanism in the brains of mice.

**Figure 6 pharmaceuticals-18-00255-f006:**
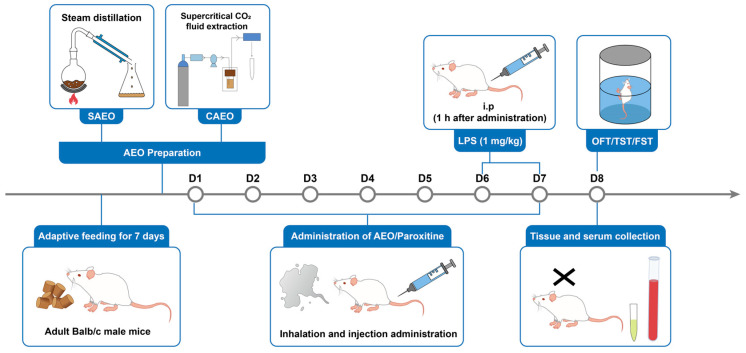
Flow chart of experimental design.

**Table 1 pharmaceuticals-18-00255-t001:** SAEO attenuates LPS model-induced depression-like behavior.

Group	Body Weight (g) (Mean ± SD)	OFT Total Distance (mm) (Mean ± SD)	FST Immobility Times (s) (Mean ± SD)	TST Immobility Times (s) (Mean ± SD)
Control	−0.07 ± 0.35	22,346.78 ± 2768.41	60.43 ± 12.42	60.43 ± 13.12
LPS-induced	−3.19 ± 0.58 ###	19,529.75 ± 3980.68	145.71 ± 11.58 ###	135.00 ± 16.90 ###
Paroxetine	−2.31 ± 0.59 *	22,703.19 ± 4097.91	75.57 ± 16.76 ***	79.75 ± 21.02 ***
SAEO Inhalation (4 μL)	−2.10 ± 0.29 ***	24,206.88 ± 4779.17	103.86 ± 14.16 ***	102.75 ± 13.47 *
SAEO Inhalation (8 μL)	−2.04 ± 0.30 ***	25,448.07 ± 5116.07	68.33 ± 11.98 ***	71.00 ± 17.47 ***
SAEO Injection	−2.02 ± 0.53 ***	23,414.93 ± 7407.16	63.17 ± 11.42 ***	65.57 ± 12.86 ***

The data are expressed as the mean ± SD (*n* = 8). ### *p* < 0.001 compared to the control group; * *p* < 0.05, *** *p* < 0.001 compared to the LPS-induced group.

**Table 2 pharmaceuticals-18-00255-t002:** Comparison of two AEO anti-LPS-induced depressive activities in mice.

Group	FST Immobility Times (s) (Mean ± SD)	TST Immobility Times (s) (Mean ± SD)
Control	59.60 ± 13.85	63.00 ± 11.81
LPS-induced	155.36 ± 30.75 ###	153.13 ± 14.46 ###
Paroxetine	62.97 ± 30.31 ***	87.32 ± 14.55 ***
SAEO Inhalation (4 μL)	81.66 ± 27.66 ***	98.61 ± 30.63 **
SAEO Inhalation (8 μL)	79.67 ± 30.49 ***	93.98 ± 25.05 ***
SAEO Injection	66.97 ± 20.40 ***	91.40 ± 19.66 ***
CAEO Inhalation (4 μL)	126.90 ± 34.02	116.75 ± 30.81
CAEO Inhalation (8 μL)	89.47 ± 41.83 **	101.21 ± 22.52 **
CAEO Injection	116.69 ± 25.29	123.64 ± 34.94

The data are expressed as the mean ± SD (*n* = 8). ### *p* < 0.001 compared to the control group; ** *p* < 0.01, *** *p* < 0.001 compared to the LPS-induced group.

**Table 3 pharmaceuticals-18-00255-t003:** SAEO reduces LPS-induced inflammation levels in serum and cortical tissues of mice.

Group	TNF-α (pg/mL) (Mean ± SD)		IL-1β (pg/mL) (Mean ± SD)		IL-6 (pg/mL) (Mean ± SD)	
	Serum	Cortex	Serum	Cortex	Serum	Cortex
Control	249.51 ± 33.49	1.06 ± 0.03	62.78 ± 4.48	0.19 ± 0.02	54.39 ± 2.43	0.27 ± 0.01
LPS-induced	406.55 ± 76.73 ###	1.59 ± 0.13 ###	101.92 ± 9.30 ###	0.29 ± 0.01 ###	81.55 ± 7.83 ###	0.33 ± 0.01 ##
Paroxetine	326.96 ± 16.38 *	1.23 ± 0.09 ***	78.02 ± 6.65 ***	0.23 ± 0.03 *	61.89 ± 3.04 ***	0.29 ± 0.01
SAEO Inhalation (4 μL)	319.37 ± 48.89 *	1.33 ± 0.06 **	79.63 ± 4.88 **	0.24 ± 0.01	67.65 ± 7.28 ***	0.29 ± 0.02 *
SAEO Inhalation (8 μL)	273.42 ± 45.38 ***	1.24 ± 0.02 ***	74.66 ± 5.80 ***	0.20 ± 0.02 **	59.26 ± 4.59 ***	0.27 ± 0.01 **
SAEO Injection	262.62 ± 16.06 ***	1.04 ± 0.06 ***	53.31 ± 5.89 ***	0.21 ± 0.01 **	59.63 ± 6.24 ***	0.25 ± 0.02 ***

The data are expressed as the mean ± SD (*n* = 4–6). ## *p* < 0.01, ### *p* < 0.001 compared to the control group; * *p* < 0.05, ** *p* < 0.01, *** *p* < 0.001 compared to the LPS-induced group.

**Table 4 pharmaceuticals-18-00255-t004:** Quantitative results of *p*-NF-κB/ NF-κB and *p*-IκB-α/ IκB-α relative protein expression levels in hippocampal tissues of Balb/c mice.

Group	*p*-NF-κB/ NF-κB Relative Protein Expression (%)	*p*-IκB-α/ IκB-α Relative Protein Expression (%)
Control	0.95 ± 0.08	1.00 ± 0.00
LPS-induced	2.09 ± 0.13 ##	2.18 ± 0.06 ###
Paroxetine	1.19 ± 0.37 *	1.39 ± 0.15 ***
SAEO Inhalation (4 μL)	1.50 ± 0.23	1.36 ± 0.17 ***
SAEO Inhalation (8 μL)	1.07 ± 0.35 **	1.08 ± 0.24 ***
SAEO Injection	1.07 ± 0.36 **	1.31 ± 0.31 ***

The data are expressed as the mean ± SD (*n* = 3–5). ## *p* < 0.01, ### *p* < 0.001 compared to the control group; * *p* < 0.05, ** *p* < 0.01, *** *p* < 0.001 compared to the LPS-induced group.

**Table 5 pharmaceuticals-18-00255-t005:** Quantitative results of BDNF/GAPDH, *p*-TrkB/GAPDH, and *p*-CREB/CREB relative protein expression levels in hippocampal tissues of Balb/c mice.

Group	BDNF/GAPDH Relative Protein Expression (%)	*p*-TrkB/GAPDH Relative Protein Expression (%)	*p*-CREB/CREB Relative Protein Expression (%)
Control	1.06 ± 0.07	1.02 ± 0.02	1.07 ± 0.04
LPS-induced	0.61 ± 0.10 ###	0.74 ± 0.11 ##	0.54 ± 0.06 ###
Paroxetine	0.92 ± 0.14 *	1.09 ± 0.10 ***	0.88 ± 0.16 *
SAEO Inhalation (4 μL)	0.78 ± 0.12	1.01 ± 0.08 **	0.65 ± 0.18
SAEO Inhalation (8 μL)	0.98 ± 0.18 **	1.11 ± 0.12 ***	0.92 ± 0.08 *
SAEO Injection	0.95 ± 0.10 **	1.15 ± 0.14 ***	1.01 ± 0.12 **

The data are expressed as the mean ± SD (*n* = 3–5). ## *p* < 0.01, ### *p* < 0.001 compared to the control group; * *p* < 0.05, ** *p* < 0.01, *** *p* < 0.001 compared to the LPS-induced group.

## Data Availability

The datasets and materials used and/or analyzed during the current study are available from the corresponding author upon reasonable request. The data cannot be made publicly available due to privacy concerns.

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
