# Peer review of "Antidepressant Activity of Agarwood Essential Oil: A Mechanistic Study on Inflammatory and Neuroprotective Signaling Pathways"

_pharmaceuticals, 2025, doi:10.3390/ph18020255_

Round 1
Reviewer 1 Report (Previous Reviewer 2)
Comments and Suggestions for Authors
The primary concern is the measurement of transcription factors/signaling molecules in the hippocampus; yet cytokines were measured in the cortex.
These should all be measured in both brain regions. the summary figure suggesting that events in hippocampus drive events in the cortex are not supported by the data.
Author Response
Please see the attachment.

Reviewer 2 Report (New Reviewer)
Comments and Suggestions for Authors
An interesting study and convincing findings.
My comments are as follows:
Introduction: The group's previous study - Please clarify which and how many group of studies. Did these groups work on similar parameters?
State the source of paraxetine / its brand name
State the brand name of electronic aromatherapy oven
For each Figure /Table, explain each abbreviation used in the caption.
Figure 6 - what is (1 hr after administration)
Figure 6 - unclear symbol used. What does an X placed on top of a rat mean?
Provide the brand name, district and country) of all chemicals and equipment used.
I need of reference for the protocol of behavioural tests.
It is unclear what are the positive drugs.
Author Response
Please see the attachment.

Reviewer 3 Report (New Reviewer)
Comments and Suggestions for Authors
1. Your abstract should have a recommendation, suggesting the future of your research.
2. There are some aspects of results eg
"AEO and positive drugs were administered for 7 consecutive days. Except for the control group which was injected with the corresponding volume of normal saline, all mice were intraperitoneally injected with 1 mg·kg-1 LPS 1 h after the administration of AEO and positive drugs starting from the 5th day. Two to six hours after LPS intraperitoneal injection, mice developed acute and severe symptoms, such as generalized shivering, elevated body temperature, reduced feeding, massive excretion of unformed feces, and rapid weight loss, consistent with documented findings in the literature. After 24 h, mice gradually recovered their voluntary activity behavior; however, during Tail suspension test (TST) and Forced swimming test (FST), there was an increase in immobility and signs of depression [33]" which should be captured under materials and methods. All aspects of the results that dealt with the description of processes should be moved to methods.
3. The results should have numbers (figures) to support them, such as p values, SEM or SD, f, df, etc. Hence, they should be presented in the language of statistics.
4. Further, multiple formats of results presentations are not allowed. You have to present your results in either tables or graphs but not both for the same result.
5. Your results section contains many literature reviews and discussions, which should be moved to the discussion section, which is meant for that purpose.
6. Your discussion section should start by highlighting your study's main findings, not a literature review.

ok
Author Response
Please see the attachment.

Reviewer 4 Report (New Reviewer)
Comments and Suggestions for Authors
1. The introduction provides a comprehensive overview, but it lacks a clear statement of the research gap and hypothesis. Please explicitly outline how this study advances current knowledge in the antidepressant activity of agarwood essential oil (AEO).
2. The experimental design needs clarification. Provide detailed justifications for selecting specific doses of AEO and the rationale for comparing SAEO and CAEO extraction methods.
3. The statistical analysis section should elaborate on the specific tests used for multiple comparisons and include a statement about how assumptions of these tests were verified.
4. Ensure that all figures are self-explanatory with clear legends. For instance, Figure 1 lacks sufficient detail about statistical comparisons between groups. Consider including annotations directly on the graphs.
5. Some tables (e.g., Tables 1 and 2) do not include a clear description of the sample size (n). Add this information for clarity.
6. The comparison of SAEO and CAEO antidepressant effects lacks in-depth discussion on the chemical composition differences. Include a quantitative analysis of key components like sesquiterpenes and their potential contributions to the observed effects.
7. Explain why only specific doses of AEO were compared and not a wider range, especially for CAEO.
8. The discussion should better contextualize findings within the existing literature. Expand on the potential molecular mechanisms and compare these results to similar studies on essential oils.
Address limitations, such as the focus on male mice only, which may impact generalizability to other populations.
9. The conclusions appear overly broad given the study's limitations. Consider refining this section to reflect the specific contributions and the need for further validation in human studies.
10. Some citations are outdated or incomplete (e.g., [1]-[4]). Ensure all references are up to date and adhere to journal guidelines.
11. Include a detailed description of how animal welfare was ensured during the experiments.
Comments on the Quality of English LanguageRevise the manuscript for grammatical errors and improve readability. Certain sections, such as the abstract and results, can benefit from more concise phrasing.
Round 2
Reviewer 4 Report (New Reviewer)
Comments and Suggestions for Authors
-
This manuscript is a resubmission of an earlier submission. The following is a list of the peer review reports and author responses from that submission.
Round 1
Reviewer 1 Report
Comments and Suggestions for Authors
The present manuscript titled “Antidepressant activity of agarwood essential oil: a mechanistic study on inflammatory and neuroprotective signaling pathways” led by Zhang et al shows the antidepressant activity of EAO against LPS-induced depression. The findings can be acceptable with the following modifications.
1. Limbic system (Thalamus, hippocampus, amygdala, hypothalamus, and etc) plays a major role in Depression. The authors have considered only the hippocampus for the present study. Please explain specific reasons for selecting the hippocampus instead whole brain?
2. The introduction is well-written. However, authors should include about BDNF/TrkB/CREB pathway role in depression. Also clearly explain why the have authors targeted this pathway.
3. Provide animal ethics approval number.
4. Provide references for OFT, FST, and TST
5. Section 4.2.2: ELISA tests were performed in cortical tissues, whereas western blot analysis was performed with the hippocampus. Please explain the reasons. The following sentence “brain supernatant samples” may not be appropriate as the authors have not considered the whole brain. Therefore, correct it.
6. Authors found that SAEO shows better antidepressant activity than CAEO. Please explain the possible reasons pertaining to their chemical composition.
7. Error! Reference source not found. Please remove this kind of words.
8. Authors have characterized the Agarwood extract with GC-MS. However, the data is missing in the manuscript.
9. 2 results: TST(?) and FST(?). Please indicate the meanings of question mark. Author should carefully correct these errors. Further, it can be assumed that these question marks represent figure numbers. If yes, please correct it.
10. Figure 1 and 2 can be merged.
11. There was no * on the figure bars in Figure 2. Therefore, authors should remove the p-value from its figure legend
12. Authors are claiming that SAEO is a better antidepressant than CAEO. Authors have claimed it is based on their immobility periods. However, authors have not measured the inflammatory markers with CAEO, which authors could have done to confirm their statement. Moreover, CAEO also shows antidepressant activity against LPS-induced inflammation-mediated depression indicating that its antidepressant activities are due to its anti-inflammatory activities. Therefore, the authors should clearly state the reasons for antidepressant activity for CAEO.
13. Authors have used Paroxetine as a standard compound. However, Paroxetine exhibits antidepressants because of its SSRI actions. Whereas authors have used an inflammation model to induce depression, but they have used SSRI for comparison of the antidepressant potential of AEO. Is it correct? Authors could have also used a notable anti-inflammatory compound to test the potential of AEO.
14. Authors could have estimated neurotransmitters along with behavioral measurements to confirm the development of depression.
Reviewer 2 Report
Comments and Suggestions for Authors
The premise of this study is very interesting. However, there are numerous concerns that must be addressed.
1. The experimental design and findings reported are not sufficient to directly compare SAEO and CAEO. For instance, the components of each extract are not defined thus it is difficult to interpret what exactly is being compared/tested. Also, only select data we presented for CAEO. Fig. is the only place SAEO and CAEO are compared and data indicate 8 ul has similar effect. Although it is unclear what this truly means given the differences in components seemingly inhaled.
2. The Results section should include statistical data for the various differences reported.
3. Methods indicate SEM reported, whereas, Fig. Legends state SD.
4. It is unclear which post-hoc pairwise test was used.
5. The design should include a Control Vehicle Inhalation group
6. Cytokines were measured in cortex, whereas upstream signaling factors were measured in hippocampus. The rationale for assessing these factors in different tissues is unclear; and interpretation of the findings is not sufficient.
7. Fig 6 is not informative. The arrows coming off LPS suggest LPS is interacting with TrkB, yet there is no data to support this interaction. The figure suggests all events occur in the same tissue yet data reflect cytokines in the cortex and NFkB, IkB, CREB etc in the hippocampus. The figure suggests SAEO site of action is LPS, yet it is unclear which finding suggests that SAEO interacts with LPS.
Round 2
Reviewer 1 Report
Comments and Suggestions for Authors
Authors have addressed all the issues satisfactorily.
Authors can cite the following work, which describing the importance of natural products in managing the depression and other neurological disorders in the introduction. https://doi.org/10.1002/ptr.8122.
Reviewer 2 Report
Comments and Suggestions for Authors
The attempt to address the concerns is appreciated, however, only some of the concerns were adequately addressed. The following issues remain:
1. Why are only select CAEO data presented?
2. Natural breathing is not the same as inhaling vehicle
3. Regarding measuring cytokines in the cortex and transcription factors in the hippocampus is not justified by the literature presented.
4. The summary figure remains concerning; it seems to suggest that NFkB expression in the hippocampus is modulating cytokine expression in the cortex. How does this occur? Which data support this?
Round 3
Reviewer 2 Report
Comments and Suggestions for Authors
The efforts of the authors are appreciated, however, only Response 1 regarding limited data for CAEO is satisfactory.
Regarding responses 2-4, these do not fully address the concerns.
#2- Vehicle inhalation is the appropriate control for drug inhalation, and it is still not included.
#3 and #4- Mechanistically, it does not make sense to try and connect cytokine expression in one tissue (i.e, cortex) with upstream transcription factors in another tissue (i.e., hippocampus). These events in each tissue may in fact be instrumental in AEO-mediated effects, but the data do not support that these responses are "interconnected".